# Isosinensetin Stimulates Glucagon-like Peptide-1 Secretion via Activation of hTAS2R50 and the G_βγ_-Mediated Signaling Pathway

**DOI:** 10.3390/ijms24043682

**Published:** 2023-02-12

**Authors:** Seung-Hyeon Lee, Hyun Min Ko, Wona Jee, Hyungsuk Kim, Won-Seok Chung, Hyeung-Jin Jang

**Affiliations:** 1College of Korean Medicine, Kyung Hee University, 26, Kyungheedae-ro, Dongdaemun-gu, Seoul 02447, Republic of Korea; 2Department of Science in Korean Medicine, Graduate School, Kyung Hee University, Seoul 02447, Republic of Korea

**Keywords:** bitter taste receptor, taste receptor type 2 member 50, hTAS2R50, isosinensetin, glucagon-like peptide-1, enteroendocrine L cell

## Abstract

Bitter taste receptors (TAS2Rs) are G protein-coupled receptors localized in the taste buds of the tongue. They may also be present in non-lingual organs, including the brain, lung, kidney, and gastrointestinal (GI) tract. Recent studies on bitter taste receptor functions have suggested TAS2Rs as potential therapeutic targets. The human bitter taste receptor subtype hTAS2R50 responds to its agonist isosinensetin (ISS). Here, we demonstrated that, unlike other TAS2R agonists, isosinensetin activated hTAS2R50 as well as increased Glucagon-like peptide 1 (GLP-1) secretion through the G_βγ_-mediated pathway in NCI-H716 cells. To confirm this mechanism, we showed that ISS increased intracellular Ca^2+^ and was suppressed by the IP_3_R inhibitor 2-APB as well as the PLC inhibitor U73122, suggesting that TAS2Rs alters the physiological state of enteroendocrine L cells in a PLC-dependent manner. Furthermore, we demonstrated that ISS upregulated proglucagon mRNA and stimulated GLP-1 secretion. ISS-mediated GLP-1 secretion was suppressed in response to small interfering RNA-mediated silencing of G_α_-gust and hTAS2R50 as well as 2-APB and U73122. Our findings improved the understanding of how ISS modulates GLP-1 secretion and indicates the possibility of using ISS as a therapeutic agent in the treatment of diabetes mellitus.

## 1. Introduction

Taste, or gustation, is associated with food preference and intake and is mediated by distinctive taste receptors present in the taste buds: sweet, umami, bitter, sour, and salty [1]. Taste receptors and taste signal transduction cascades are localized on either the tongue or in non-lingual organs, including the brain [2], lung [3], gastrointestinal tract [4,5,6,7], and colon [8]. Bitterness causes an innate aversion to the digestion of harmful substances [9]. On the other hand, bitter compounds also influence diet, metabolic processes, and homeostasis [10,11]. In this regard, bitter taste receptors, which are gut-expressed chemoreceptors, were found in enteroendocrine L cells, which secrete glucagon-like peptide-1 (GLP-1) [5]. Meanwhile, the association between how enteroendocrine L cells sense bitterness and GLP-1 secretion is not well understood.

Bitter taste receptors (also known as taste family 2 receptors, or TAS2Rs) belong to the superfamily of seven-transmembrane G-protein-coupled receptors (GPCRs), which are composed of trimers α, β, and γ-subunits [12]. By interacting with specific ligands, the G-protein α-subunit of gustducin (Gα gust) stimulates phosphodiesterase (PDE) to reduce intracellular cAMP levels, or the G-protein βγ subunit (G_βγ_) stimulates phospholipase Cβ2 (PLCβ2) to release Ca^2+^ from intracellular stores through the inositol 1,4,5-triphosphate (IP_3_)/diacylglycerol (DAG) pathway [13,14]. As a result, the ion gradient alters the membrane potential, resulting in the secretion of vesicles, which contain incretin hormones such as GLP-1.

GLP-1 derived from intestinal enteroendocrine L cells is an important gut incretin hormone encoded by the proglucagon gene [15]. Through the taste receptor transduction cascade, plasma levels of GLP-1 rapidly increase within minutes of nutrient ingestion and reach pancreatic β cells to secrete insulin as well as promote β-cell proliferation and act on the central nervous system to reduce appetite [15,16,17]. However, secreted GLP-1 is rapidly removed by the enzyme dipeptidyl peptidase-4 (DPP-4). Therefore, DPP-4 inhibitors or GLP-1 receptor agonists, which exploit the insulinotropic effect of GLP-1, are prescribed for treating type 2 diabetes mellitus [18].

Recent studies have reported that the expression of hTAS2Rs plays a crucial role in the regulation of physiological metabolism and disease etiology [19]. In a recent study, hTAS2R38 was reported to be associated with food consumption and obesity risk [20], and hTAS2R 10, 14, and 31 were associated with obstructive pulmonary disease [21,22,23]. Furthermore, the activation of hTAS2R38 has antibacterial effects in sinonasal epithelial cells [24]. Several experiments related to taste signal transduction cascades have been conducted to study the physiological functions of TAS2Rs in vitro and in vivo [25,26,27,28,29,30], and agonists that stimulate TAS2Rs in response to bitter compounds have been profiled [31,32]. However, the physiological functions of other hTAS2Rs have not been well elucidated.

In our previous study, isosinensetin (ISS, [C_20_H_20_O_7_]), a major active constituent of *Bupleurum falcatum* L. (BF), was identified through DART-MS analysis [33]. BF is traditionally used for treating type 2 diabetes mellitus (T2DM) in Korea [34]. Our previous study demonstrated that treatment with BF alleviates hyperglycemia through the G_βγ_ pathway-mediated regulation of GLP-1 secretion [33]. Isosinensetin, an isoform of sinensetin, belongs to the family of flavonoids with anti-inflammatory and antioxidant effects [35,36]. However, the anti-diabetic effects of isosinensetin have not been well elucidated.

In this study, we investigated the expression of bitter taste receptors by treating NCI-H716 cells with ISS. Moreover, we studied whether ISS stimulates GLP-1 secretion through activation of hTAS2R50. The current study provides insights into the mechanism underlying ISS-mediated GLP-1 secretion and suggests ISS as a potential therapeutic agent against diabetes mellitus.

## 2. Results

### 2.1. ISS Activates NCI-H716 Cells via the Phospholipase C Pathway

One of the representative flavones (backbone structure shown in Figure 1A) was tested, isosinensetin (Figure 1B) from *Bupleurum falcatum* L. We measured the cytotoxicity of ISS in NCI-H716 cells using MTT assay. No cytotoxicity was induced in NCI-H716 cells exposed to ISS (up to 80 μM) for 1 h. We sought to determine whether ISS induces an increase in the intracellular calcium concentration in enteroendocrine cells through the PLC β2 pathway. We used U73122, a PLC β2 inhibitor, and 2-APB, to block the release of the secondary messenger IP_3_. As a result, the intracellular calcium concentration increased against increasing doses of ISS (Figure 2A). Meanwhile, the intracellular calcium concentration was reduced in NCI-H716 cells after pre-treatment with U73122 (10 μM) or 2-APB (10 μM) compared to the ISS-treated group (Figure 2B,C). These results indicate that ISS induces an increase in the intracellular calcium concentration through PLC β2 signal transduction cascades.

### 2.2. Profiling the Expression of hTAS2R Genes

To investigate the expression of hTAS2R genes in differentiated NCI-H716 cells in response to 80 μM of ISS, we examined the expression levels of 25 human bitter taste receptor genes and the Gαgust-encoding gene (GNAT3) using real-time quantitative PCR (Figure 3). The mRNA expression levels of the 25 human bitter taste receptors were normalized to that of GAPDH. The results indicated that, except for hTAS2R1 and hTAS2R16, the relative abundance of the taste receptors, namely, hTAS2R50, was much higher than those of other receptors. Based on this result, we hypothesized that the high expression levels of hTAS2R50 in response to ISS might be related to GLP-1 secretion.

### 2.3. hTAS2R50 Knockdown Suppresses the Isosinensetin-Mediated Phospholipase C Pathway

To confirm that hTAS2R50 was involved in ISS-induced GLP-1 secretion, we knocked down hTAS2R50 through siRNA transfection in NCI-H716 cells. The results showed that ISS treatment increased hTAS2R50 by approximately 2-fold compared to the non-treated group and decreased the expression of TAS2R50 in si-hTAS2R50 compared to that in both ISS-treated groups. (Figure 4A). In previous studies, it was reported that GLP-1 secretion is positively correlated with α-gustducin and TAS2Rs expression as well as an increase in the intracellular calcium concentration [5,30]. To investigate the difference in the intracellular calcium concentration among hTAS2Rs, we transfected GNAT3 and hTAS2R50. As a result, a decreased intracellular calcium concentration was observed in both si-GNAT3 and hTAS2R50 compared to the si-control group (Figure 4B). Likewise, a similar pattern of results was obtained when experiments were performed with different concentrations of ISS (Figure 4C). Therefore, our results suggest that hTAS2R50 is involved in GLP-1 secretion via PLC β2 signal transduction cascades.

### 2.4. Isosinensetin Stimulates the Secretion of Proglucagon-Encoding GLP-1

Real-time PCR analysis of proglucagon-encoding glucagon, GLP-1 and GLP-2 showed that the mRNA expression of proglucagon was significantly upregulated in response to ISS treatment in a dose-dependent manner (Figure 5A). In addition, compared to the siRNA control group, the expression of proglucagon was significantly decreased in both the si-Gαgust and si-hTAS2R50 groups (Figure 5B). These results suggest that Gαgust expression correlates with the hTAS2R50 signaling pathway, and the PLC β2 pathway correlates with GLP-1 secretion.

### 2.5. Isosinensetin Stimulates hTAS2R50 and Induces GLP-1 Secretion

GLP-1 secretion in NCI-H716 cells was confirmed using a GLP-1 ELISA kit. The results showed that the level of GLP-1 increased in ISS-treated cells in a concentration-dependent manner compared to that in the non-treated cells (Figure 6A). To investigate the relationship between ISS-activated hTAS2R50 and GLP-1 secretion, we blocked the expression of Gαgust and hTAS2R50 by transfecting siRNAs targeting the signaling sequences (Figure 6B). Our previous study demonstrated that both Gαgust and hTAS2Rs are correlated with GLP-1 secretion in differentiated NCI-H716 cells [5]. As a result, the level of GLP-1 decreased not only in si-Gαgust but also in the si-hTAS2R50 group compared to the vehicle (no transfection) and siRNA control groups. To further confirm the relationship between the Gβγ-mediated signaling pathway and GLP-1 secretion, we used U73122 and 2-APB (Figure 6C,D). GLP-1 levels in the U73122 and 2-APB-treated groups decreased but were higher compared to the GLP-1 level of the non-treated group because inhibitors of bitter taste are not 100% effective. Thus, we demonstrated that ISS-activated TAS2R50 was associated with GLP-1 secretion through Gβγ-mediated signal transduction cascades.

## 3. Discussion

GLP-1, which was initially identified in the gut and is produced by L cells, stimulates glucose-mediated insulin secretion by pancreatic beta cells through blood circulation [5,16,18]. Moreover, secreted GLP-1 has been identified as a novel regulator of glucose homeostasis [10]. It is well known that bitter-tasting pharmaceuticals affect glucose homeostasis, but the relationship between bitter substances and TAS2R-stimulated GLP-1 secretion has not been reported [27,29,37]. In our previous study, treatment with denatonium benzoate (DB) and *Gentiana scabra*, a bitter tastant, led to α-gustducin and taste receptor-mediated GLP-1 release in *db/db* mice [25,30]. These experiments indicated that dietary intake regulates TAS2R expression and suggested that TAS2R is a viable pharmaceutical target for the treatment of diabetes mellitus. In this study, we confirmed that ISS induces GLP-1 secretion through the G_βγ_-mediated pathway, and ISS-mediated activation of TAS2Rs results in subsequent GLP-1 secretion.

Some representative flavonoids used in traditional herbal medicines have been reported to have anti-inflammatory, anti-oxidant, and anti-obesity effects [38,39]. Studies have also reported that bitter-tasting flavonoids stimulate the bitter taste receptors [40]. In this study, we used one of the representative flavones (backbone structure shown in Figure 1A), isosinensetin (ISS) obtained from *Bupleurum falcatum* L. (BF), which is structurally similar to flavones to stimulate hTAS2Rs (Figure 1B).

Based on the MTT assay results, we used ISS to investigate whether ISS activates TAS2Rs and stimulates Ca^2+^ increase in NCI-H716 cells. DB was used as a positive control for comparison purposes. The results revealed an increased flux of Ca^2+^ into the cytosol in response to ISS (up to 80 μM, Figure 2A, and 10 mM DB). Likewise, using pathway inhibitors such as PLC inhibitor U73122 and IP_3_ receptor antagonist 2-APB, we also measured Ca^2+^ levels in the ISS-treated NCI-H716 cells (Figure 2B,C). The results showed that the intracellular calcium concentration was sufficiently reduced by pre-treatment with U73122 or 2-APB compared to ISS treatment. Therefore, we demonstrated that ISS stimulates PLC β2 through taste signal transduction cascades.

Recently, some studies have reported that the expression of bitter taste receptors plays a crucial role in the regulation of physiological metabolism and disease etiology [19,41]. Numerous studies have reported the role of taste signal transduction cascades in the physiological functions of TAS2Rs in vitro and in vivo [25,26,27,28,29,30,42]. In addition, studies of profiling of agonists stimulating hTAS2Rs with bitter compounds have been reported [31,32]. However, the function of TAS2Rs in humans is not well understood. In this study, to visualize the expression of the human TAS2R gene in NCI-H716 cells, we decided to perform quantitative real-time PCR experiments. As shown in Figure 3, the amplification of 25 bitter taste receptors from NCI-H716 cDNA revealed that all transcripts were present in NCI-H716 cells under ISS treatment. As a result, except for hTAS2R1 and hTAS2R16, the mRNA expression of taste receptors such as hTAS2R8, hTAS2R45, and hTAS2R50 was much higher compared with that of other receptors. In particular, hTAS2R50 was most highly expressed. In this regard, previous studies have reported that two natural bitter terpenoids, andrographolide and amarogentin, selectively activate hTAS2R50 [43]. Therefore, we demonstrated that ISS selectively activated hTAS2R50 and hypothesized that ISS-activated hTAS2R50 was associated with the PLC β2 pathway through taste signal transduction cascades in NCI-H716 cells.

To confirm that hTAS2R50 was involved in the ISS-induced intracellular calcium response, we knocked down hTAS2R50 through siRNA transfection in NCI-H716 cells. As shown in Figure 4A, the ISS treatment increased the expression of hTAS2R50 by approximately 2-fold compared to a non-treated control group. On the other hand, when the expression of hTAS2R50 was suppressed using si-hTAS2R50, the expression of hTAS2R50 was lower compared to the ISS-treated groups. Based on these results, we conducted calcium imaging to indicate whether hTAS2R50 correlates with the intracellular calcium response as well as the association with the PLC β2 pathway through taste signal transduction cascades in NCI-H716 cells. Our previous study demonstrated that both Gαgust and hTAS2Rs are correlated with GLP-1 secretion in differentiated NCI-H716 cells [29]. In addition, the influx of calcium in response to ISS treatment was dose-dependently decreased in the si-Gαgust and si-hTAS2R50 group compared to the si-control group (Figure 4B,C). However, a slight decrease in the intracellular calcium concentration in both siRNA groups in response to ISS treatment suggests the correlation of these receptors for ISS as well as the existence of multiple receptors. Incretin hormones such as glucagon, GLP-1, and GLP-2 encoded by the proglucagon gene are involved in glucose homeostasis. To confirm the correlation of GLP-1 secretion, we performed real-time PCR analysis. The results showed that the mRNA expression of proglucagon was upregulated in response to ISS treatment in a concentration-dependent manner (Figure 5A). However, proglucagon expression was significantly downregulated in the siRNA groups (Figure 5B). Therefore, this result suggests hTAS2R50 and Gαgust are associated with proglucagon-encoding GLP-1 hormone. Based on this result, we further confirmed that the GLP-1 levels significantly increased against increasing doses of ISS in NCI-H716 cells (Figure 6A). In addition, the level of GLP-1 was attenuated in not only Gαgust but also in the hTAS2R50 siRNA group compared to the siRNA control groups (Figure 6B). To further confirm the relationship between the G_βγ_-mediated signaling pathway and GLP-1 secretion, we used U73122 and 2-APB (Figure 6C,D). As a result, the GLP-1 levels in the U73122- and 2-APB-treated groups decreased compared to the ISS group. Thus, we demonstrated that ISS-activated hTAS2R50 was associated with GLP-1 secretion through G_βγ_-mediated signaling.

In this study, we demonstrated that ISS stimulates GLP-1 secretion by activating hTAS2R50 and Gαgust-mediated signal transduction cascades in NCI-H716 cells. Although multiple receptors respond to ISS treatment, for the first time, we revealed that ISS stimulates GLP-1 secretion through the activation of the human bitter taste receptor hTAS2R50 as well as G_βγ_-mediated signal transduction cascades in NCI-H716 cells. However, one of the major limitations of this study is that the results are based on NCI-H716 cells only and may not be generalizable to other cell lines or in vivo. Further research using a wider range of cell lines or animal models is needed to fully understand the efficacy and potential limitations of the treatment. Our results improve the understanding of the mechanism modulating GLP-1 secretion and suggest the possibility of using ISS as a therapeutic agent in the treatment of T2DM.

## 4. Materials and Methods

### 4.1. Chemicals

Isosinensetin (ISS) was purchased from ChemFaces (Wuhan ChemFaces Biochemical Co., Ltd., Wuhan, China). 2-APB (10 μM) and U73122 (10 μM) were purchased from Sigma–Aldrich (St. Louis, MO, USA). MTT (3-(4,5-dimethyl-2-thiazolyl)-2,5-diphenyl-[2H]-tetrazolium bromide) was obtained from Invitrogen (Waltham, MA, USA).

### 4.2. Cell Culture

Human enteroendocrine NCI-H716 cells (ATCC^®^ CCL-251) were obtained from ATCC (American Type Culture Collection, Manassas, VA, USA). NCI-H716 cells were cultured in RPMI 1640 (Corning Inc., New York, NY, USA) supplemented with 10% fetal bovine serum (FBS; Gibco, Grand Island, NY, USA) and an antibiotic–antimycotic solution (ABAM; Corning Inc., NY, USA) at 37 °C under 5% CO_2_. To perform endocrine differentiation, NCI-H716 cells (1 × 10^6^ cells/well) were seeded onto 12-well plates coated with Matrigel (BD Bioscience, Bedford, MA, USA) and cultured in high-glucose DMEM (Corning Inc., NY, USA) supplemented with 10% FBS and ABAM. After 48 h, the medium was replaced with low-glucose DMEM and further incubated for 16 h. NCI-H716 cells expressed several neuroendocrine markers, including chromogranin A.

### 4.3. MTT Assay

NCI-H716 cells (1 × 10^4^ cells/well) were seeded onto pre-coated 96-well plates and incubated to induce the differentiation of endocrine cells for 48 h. Cells were treated with ISS (25, 50, 80, 100, and 200 μM) for 1 h to confirm the cytotoxicity of ISS. MTT was added to the wells at a final concentration of 0.5 mg/mL and incubated for 2 h. After removing the medium, formazan crystals were dissolved in DMSO and absorbance was recorded on a microplate reader (Bio-Rad Hercules, Hercules, CA, USA) at 540 nm.

### 4.4. Calcium Imaging

Differentiated NCI-H716 cells were seeded onto a clear-bottom 96-well black plates (Corning, Tewksbury, MA, USA) pre-coated with Matrigel. Before the experiment, the medium was replaced with PBS and incubated for 30 min with fura-2 AM dye (final concentration 1 μM) as described previously. Then, the medium was replaced with U73122 (10 μM, Sigma–Aldrich) and 2-aminoethoxydiphenyl borate (2-APB; 10 μM, Sigma–Aldrich) in fresh PBS for 30 min. Cytosolic free calcium [Ca^2+^]i was observed using a Nikon Eclipse TS 100 fluorescence imaging system (Nikon Instrument Inc., Melville, NY, USA). Alternating excitation wavelengths of 340 and 380 nm were delivered from a rotating wheel filter, and images were analyzed using *InCyt* Im2 software (Intracellular Imaging; Cincinnati, OH, USA).

### 4.5. Real-Time PCR

NCI-H716 cells (1 × 10^6^ cells/well) were seeded onto 6-well plates pre-coated with Matrigel and differentiated with DMEM for 48 h. After starvation, fully differentiated cells were incubated with or without ISS (80 μM) for 1 h. To isolate RNA, the cells were treated with Ribo-Ex using the GeneAll Hybrid-R RNA Purification Kit according to the manufacturer’s instructions (GeneAll, Seoul, Korea). RNA was quantified using a NanoDrop spectrophotometer (Thermo Fisher Scientific, Waltham, MA, USA). The amplification of cDNA was performed as follows: 45 °C for 60 min and 95 °C for 5 min using the Maxime RT premix kit (iNtRON BIOTECHNOLOGY, Seongnam, Korea). Real-time quantitative PCR was performed using the Universal SYBR Green Master Mix (Applied Biosystems, Foster City, CA, USA) on an Applied Biosystems StepOne System (Applied Biosystems, Foster City, CA, USA). In this study, the expression levels of genes belonging to the TAS2R family were quantified relative to that of GAPDH to determine the mRNA levels. The PCR primers used in this study are listed in Table 1.

### 4.6. Western Blot

NCI-H716 cells (1 × 10^6^ cells/well) were seeded onto pre-coated 6-well plates and incubated at 37 °C under 5% CO_2_ for 48 h. After starvation, the cells were treated with ISS for 1 h. The NCI-H716 cells were lysed with cell lysis buffer (Cell Signaling, Danvers, MA, USA). Proteins were separated by SDS-PAGE (10%) and transferred to a PVDF membrane. The membrane was blocked with 3% bovine serum albumin (BSA) and incubated with primary antibodies against TAS2R50. The TAS2R50 polyclonal antibody and β-actin were purchased from Bioss Antibodies (Woburn, MA, USA) and Santa Cruz (Dallas, TX, USA), respectively. The protein expression levels were developed using an enhanced chemiluminescence (ECL) detection system and detected using an ImageQuant LAS 500 imaging system (GE Healthcare Life Sciences, New Zealand, NSW, Australia). The intensities of the individual bands were calculated using ImageJ software (NIH, New York, NY, USA).

### 4.7. siRNA Preparation and Transfection

Transfection of siRNA was performed using lipofectamine 3000 Reagent (Invitrogen, Carlsbad, CA, USA) according to the manufacturer’s protocol. The predesigned scrambled negative control siRNAs for Gαgust (product name: 1175215) and hTAS2R50 (product name: 259296-1) were purchased from Bioneer (Daejeon, Korea). NCI-H716 cells were seeded onto 24-well and 96-well black plates coated with Matrigel along with siRNA solution diluted in DMEM and incubated for 48 h. After transfection, the medium was replaced with ISS and cells were incubated for an additional 1 h.

### 4.8. GLP-1 ELISA

NCI-H716 cells (1 × 10^6^ cells/well) were seeded onto 12-well pre-coated plates and incubated for 48 h in siRNA solution diluted in DMEM to induce differentiation. After starvation, the cells were pre-treated with U73122 and 2-APB for 30 min. The medium was then exchanged with 1 mL/well DMEM containing 80 μM ISS, and the cells were incubated for 1 h. Then, the cell culture medium was centrifuged at 2000× *g* for 10 min to remove debris, and 50 μL of the supernatant was used for GLP-1 analysis. For the quantitative measurement of GLP-1 in the cell culture supernatants, a human GLP1 (7-36) enzyme-linked immunosorbent assay (ELISA) kit (ab184857) was purchased from Abcam (Cambridge, UK). The GLP-1 ELISA was performed according to the manufacturer’s protocol.

### 4.9. Statistical Analysis

All in vitro experiments were repeated at least three times. Data represent the mean ± SEM and SD of at least three independent experiments. Statistical analysis was performed using a one-tailed Mann–Whitney U test and one-way ANOVA with Bonferroni’s multiple comparison test and Dunnett’s multiple comparison test in Graph Pad Prism 8 (GraphPad software, San Diego, CA, USA). A *p*-value < 0.05 was considered statistically significant, and *p* < 0.01 and *p* < 0.001 were considered highly significant.

## Figures and Tables

**Figure 1 ijms-24-03682-f001:**
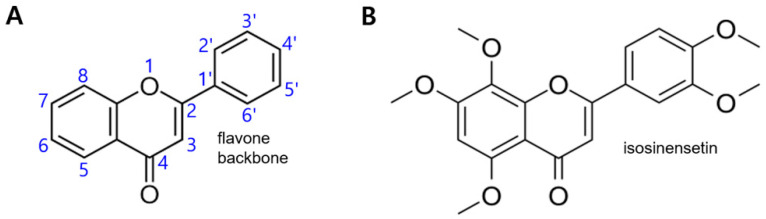
Molecular structure of flavone used in this study. (**A**) Backbone flavone structure. (**B**) Structures of isosinensetin (5,7,8,3′,4′-pentamethoxyflavone).

**Figure 2 ijms-24-03682-f002:**
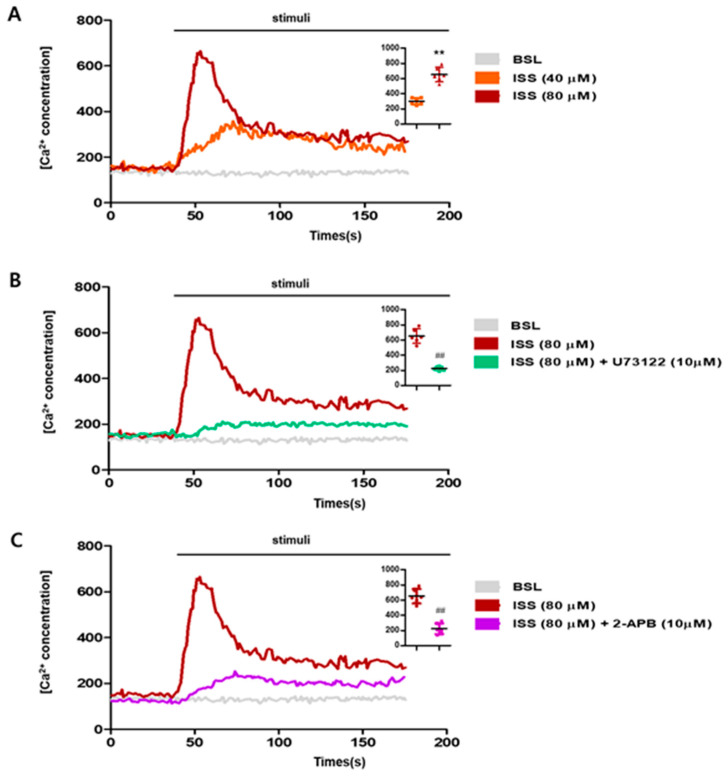
Increase in intracellular Ca^2+^ concentrations in NCI-H716 cells exposed to ISS at 40 s. Data are illustrated in representative traces as well as in scatter plots expressing the mean ± S.D. of the maximum peak of intracellular Ca^2+^ release. (**A**) Treatment with ISS dose-dependently increased the concentration of Ca^2+^ in the cytosol. (**B**) U73122 is an inhibitor of PLC-dependent processes. (**C**) 2-APB is an inhibitor of IP_3_-induced Ca^2+^ release. ISS-treated cells exposed to these two inhibitors showed fewer changes in the concentration of Ca^2+^ compared to ISS (80 μM). (**A**) Significance was determined by an unpaired *t*-test (one-tailed), ** *p*< 0.01 vs. ISS 40 μM. (**B**,**C**) Significance was determined by an unpaired *t*-test (one-tailed), ^##^
*p* < 0.01 vs. ISS 80 μM. Data are shown as the mean ± SD, with individual results, from independent experimental replicates, n = 6.

**Figure 3 ijms-24-03682-f003:**
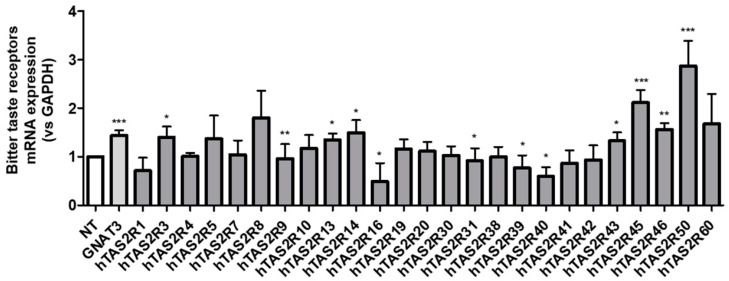
mRNA expression profiling of the hTAS2Rs in the differentiated NCI-H716 cells treated with ISS (80 μM) for 1 h. of ISS. The expression levels of all 25 hTAS2Rs and Gα-gustducin (GNAT3) were normalized to that of GAPDH. Experiments were performed in triplicate and repeated three times for each condition. Statistical analysis was conducted using an unpaired *t*-test (one-tailed). Values are expressed as the mean ± SEM; *** *p* < 0.001, ** *p* < 0.01, * *p* < 0.05 vs. NT; n = 6.

**Figure 4 ijms-24-03682-f004:**
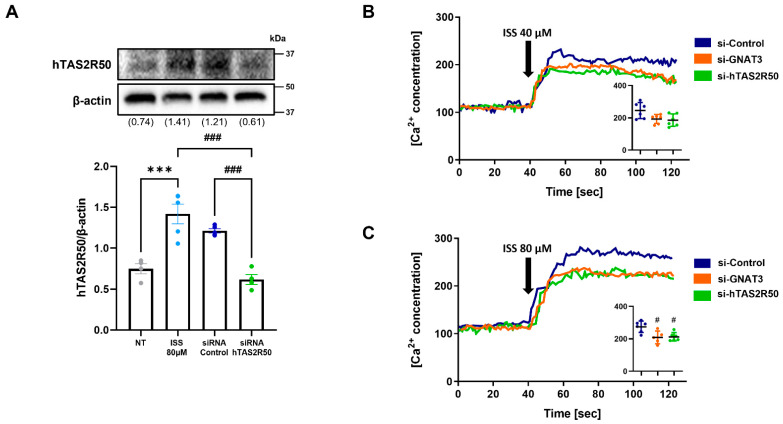
ISS-activated Ca^2+^ responses are blocked by siRNAs directed against hTAS2R50 and GNAT3. (**A**) Representative Western blot showing the extent of hTAS2R50 knockdown in NCI-H716 cells. (**B**,**C**) The effect of hTAS2R50 and GNAT3 knockdown on ISS-induced intracellular Ca^2+^ responses against increasing doses of ISS. Data are illustrated in representative traces as well as in scatter plots expressing the mean ± S.D. of the maximum peak of intracellular Ca^2+^ release. (**A**) Statistical analysis was conducted by a one-way ANOVA multiple comparison test. Data are shown as the mean ± SD, with individual results, *** *p* < 0.001, ^###^
*p* < 0.001; n = 4). (**B**) Significance was determined by one-way ANOVA and Dunnett’s multiple comparison test, but no significant difference was found. (**C**) Significance was determined by one-way ANOVA and Dunnett’s multiple comparison test, ^#^
*p* < 0.05 vs. si-control). Data are shown as the mean ± SD, with individual results, from 6 independent experiments.

**Figure 5 ijms-24-03682-f005:**
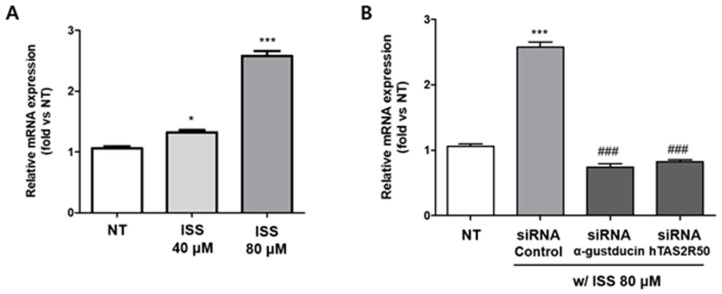
mRNA expression of the GLP-1 precursor proglucagon after ISS treatment for 1 h. (**A**) The expression of proglucagon increased against increasing doses of ISS Significance was determined by an unpaired *t*-test (one-tailed), *** *p* < 0.001, * *p* < 0.05 vs. NT. (**B**) The mRNA expression of proglucagon was significantly decreased in ISS-treated NCI-H716 cells with siRNA-mediated knockdown of GNAT3 and hTAS2R50. Statistical analysis was conducted by one-way ANOVA with Bonferroni’s multiple comparison test. Values are expressed as the mean ± SEM; *** *p* < 0.001, * *p* < 0.05 vs. NT, ^###^
*p* < 0.001, vs. siRNA control, n = 6.

**Figure 6 ijms-24-03682-f006:**
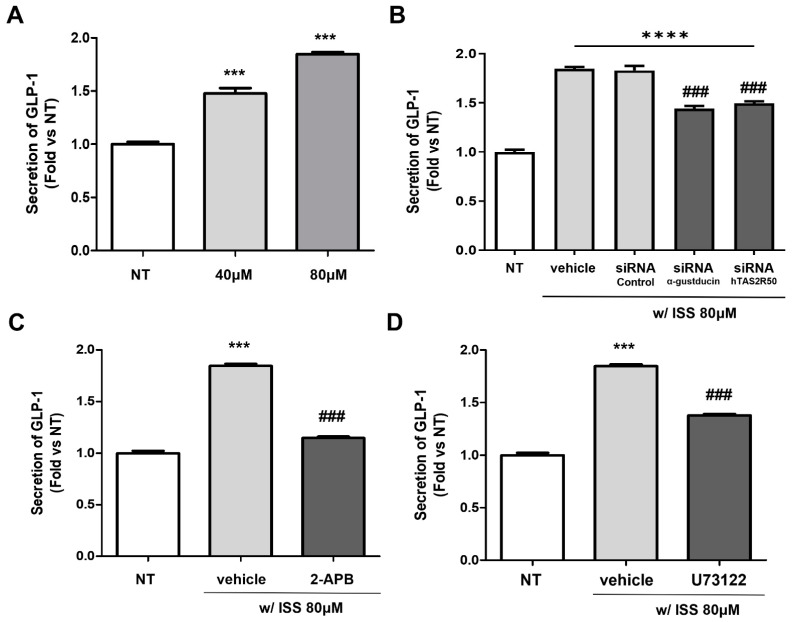
ISS-induced GLP-1 secretion in NCI-H716 cells. (**A**) Secretion of GLP-1 from ISS-treated NCI-H716 cells against increasing doses of ISS. Significance was determined by an unpaired *t*-test (one-tailed), *** *p* < 0.001 vs. NT. (**B**) siRNA-mediated decrease in GLP-1 secretion. Statistical analysis was conducted by one-way ANOVA with Bonferroni’s multiple comparison test, *** *p* < 0.001 vs. NT, **** *p* < 0.0001 vs. NT, ^###^
*p* < 0.001 vs. siRNA control. (**C**) IP_3_R inhibitor (**D**) Inhibition of PLC-dependent processes. Significance was determined by an unpaired *t*-test (one-tailed), *** *p* < 0.001 vs. NT, ^###^
*p* < 0.001 vs. vehicle. Values are expressed as the mean ± SEM, n = 4.

**Table 1 ijms-24-03682-t001:** Primers for 25 hTAS2Rs, α-gustducin, and proglucagon.

	Forward	Reverse
hTAS2R1	GCCCCATGCCTTTATTGTTA	GAGGTGAGACTCTAGCATTT
hTAS2R3	AGTGAGGAGTTCTATGTAT	TGACCAGAATCCCAGTGTG
hTAS2R4	GTGGTTTCTTTGGTCTTGAG	CCTACATGAGCTTCCGTCTG
hTAS2R5	TCGCTTGAAGACAGATTACG	GAGAAATTCAACCACTGCCA
hTAS2R7	ACTCCATCAACTAGA	ACCCAGTCCATGCAGTTAC
hTAS2R8	TTAACCTGTTTGCAATTGTC	TGGCTCTCATGAACTTCT
hTAS2R9	CTGTCATTTTCCCATCAAGC	GGGTCTCTATGGAACAAAAG
hTAS2R10	CATTTCCCTTTGGAGACACA	ATGAGCTTCTGTGTTGGAGT
hTAS2R13	AGGAGCAGAAAAAGGAGAAG	GTGAAGATACTCGGCAGCAGGG
hTAS2R14	CCTCACTGCTTTGGCAATCT	ACACACACCAGCTTCCGAAT
hTAS2R16	CCAGGAAGACACTTTGGAGT	TGACTTGCAGCCATTCTCTCTG
hTAS2R19	CGAACCATTTCAGCATGTGG	CCCCAACAGTATCACCAGAA
hTAS2R20	AGATGCGACCAAAAGAAATT	CACCTGCCACAAAACTGAAA
hTAS2R30	TTCAGCTATCCTTCAACCCA	GCCCCTCTTGTGAATCTATG
hTAS2R31	CAGCACCAAGGTCCACATAA	GTAAACGGCACATAACAAGA
hTAS2R38	TGGCAACCAGGTCTTTAGAT	ACTCCAGGACTGAAATGAAC
hTAS2R39	ATTACTGGATTGATACCCTGGC	TGTTTTCTTAGTGGAGTTGGAGG
hTAS2R40	TGCCGGCCACTCAGTACAA	ACCGCTTCCAGGCTCTTCTC
hTAS2R41	CCATGCAGAACGACTTTTAC	TTGAGGTTGCTGAAGATGAG
hTAS2R42	ACTGGTAAACTGCTCTGAAGG	ATGTGAAGCAAGTCCCACTAG
hTAS2R43	GACCACGAACCCACCTG	ACAAATGTAACCACTACCAG
hTAS2R45	CCTTTGCTGACCAAATTGTCACT	TAATAATAACACCCAGAGCAAACCAA
hTAS2R46	GACCACGAACCCACCTG	ACAAATGTAACCACTACCAG
hTAS2R50	GTTGTCATGGTTAGCAAGGC	GAGTTGAGAGTTTCAGGTCT
hTAS2R60	ACGGAGCTACTGTGAGAAAT	CGGAATCCTGAGGTTGTAAG
GAPDH	CCACTGCCGCATCCTCTTCC	CTCGTTGCCAATAGTGATGAC
GNAT3	AGAGCAAGGAGTCAGCCAAAAG	CGCTCAGCATCCTCCTGAA
Proglucagon	AATCTTGCCACCAGGGACTT	AGTGACTGGCACGAGATGTT

## Data Availability

All data presented this study are available from the corresponding author upon reasonable request.

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
