# Peer review of "Isosinensetin Stimulates Glucagon-like Peptide-1 Secretion via Activation of hTAS2R50 and the G_βγ_-Mediated Signaling Pathway"

_ijms, 2023, doi:10.3390/ijms24043682_

Round 1

Reviewer 1 Report

The study is experimentally sound, and the results are well presented and discussed. My only concern is its relevance: a concentration of up to 80 µM isosinensitin is unlikely to be reached in the lower intestinal tract for 16 hours by habitual dietary intake. Moreover, the cell line used has been established from a cecum adenocarcinoma. The authors may at least discuss these limitations of their work. Ideally, dose response data covering dietary relevant concentrations are added.

Author Response

Reviewer 1: Comments and Suggestions for Authors

The study is experimentally sound, and the results are well presented and discussed. My only concern is its relevance: a concentration of up to 80 µM isosinensetin is unlikely to be reached in the lower intestinal tract for 16 hours by habitual dietary intake. Moreover, the cell line used has been established from a cecum adenocarcinoma. The authors may at least discuss these limitations of their work. Ideally, dose response data covering dietary relevant concentrations are added.

(Response) Thanks for your comments. First, our study hypothesized that isosinensetin (ISS) stimulates GLP-1 secretion through certain human taste receptor in vitro. NCI-H716 cells is currently the only human model available for the in vitro study of GLP-1 regulation (1). The GLP-1 responses to various dietary nutrients and their respective intracellular mechanisms of action on the L-cell have been, and continue to be, intensively studied using in vitro GLP-1-secreting cell models. NCI-H716 cell line is the only line that contained dense-core granules, which are characteristic of endocrine cells. So, we used NCI-H716 cell to evaluate GLP-1 secretion in vitro. As no information are available for drug delivery to GI tract, we plan to further study using adequate mouse model to evaluate effect of ISS based on this study. If poor absorption or delivery to the intestinal tract by the oral route, we will overcome though the proper design and formulation such as encapsulation of ISS.

  • Gagnon J, Brubaker PL. NCI-H716 Cells. In: Verhoeckx K, Cotter P, López-Expósito I, et al., editors. The Impact of Food Bioactives on Health: in vitro and ex vivo models [Internet]. Cham (CH): Springer; 2015. Chapter 20.

Reviewer 2 Report

The study by Lee and colleagues is clearly described and the data back up their results.

I have only a few comments:

1. For experiments shown in Figure 4 it should be made clearer in the text and the legend that cells transfected with si-hTAS2R46 siRNA are used as controls. To me this is fine, but please can the authors explain why they don't also show results with the siRNA control.

2. Are the primers used for PCR (Table 1) intron-spanning to prevent amplification of genomic DNA? This is important since the RNA was not treated with DNAse (correct?).

3. For the GLP-1 ELISA (section 4.8), please include the volume of medium used while incubating the cells in the 12-well plates and the volume used in the ELISA.

Author Response

Reviewer 2: Comments and Suggestions for Authors

The study by Lee and colleagues is clearly described and the data back up their results.

I have only a few comments:

Point 1. For experiments shown in Figure 4 it should be made clearer in the text and the legend that cells transfected with si-hTAS2R46 siRNA are used as controls. To me this is fine, but please can the authors explain why they don't also show results with the siRNA control.

(Response) Thanks for your comments. There may have been some misunderstanding about figure 4 B and C. As you mentioned, we have modified controversial figure 4B, C and manuscript. In fact, based on the previous result (as below) that there was no difference between the si-control group and the non-treat group (no transfection), we marked it as is ISS treatment group (deep blue line in figure 4B and 4C). So, one of the reasons why we put hTAS2R46 as negative control. In this regard, the legend of figure 4 and a clear explanation were added in the revised manuscript. (Line 118-131, 224-235)

[preliminary data]

[Revised Figure 4]

Point 2. Are the primers used for PCR (Table 1) intron-spanning to prevent amplification of genomic DNA? This is important since the RNA was not treated with DNAse (correct?).

(Response) Thanks for your comments. Yes. These primers were designed to avoid amplifying any contaminating genomic DNA by spanning the introns. In addition, these primers have been used in previous studies.

Point 3. For the GLP-1 ELISA (section 4.8), please include the volume of medium used while incubating the cells in the 12-well plates and the volume used in the ELISA.

(Response) Thanks for your comments. As you mentioned, we changed section 4.8 in manuscript as describe below. Thank you for your advice.

NCI-H716 cells (1 × 106 cells/well) were seeded onto 12-well pre-coated plates and incubated for 48 h in siRNA solution diluted in DMEM medium to induce differentiation. After starvation, the cells were pre-treated with U73122 and 2-APB for 30 min. The medium was then exchanged with 1 mL/well DMEM medium containing 80 μM ISS and the cells were incubated for 1 h. And then the cell culture media was centrifuged at 2,000x g for 10 minutes to remove debris. 50 μL of supernatants were used for GLP-1 analysis. For the quantitative measurement of GLP-1 in cell culture supernatants, Human GLP1 (7-36) enzyme-linked immunosorbent assay (ELISA) kit (ab184857) was purchased from Abcam (Cambridge, UK). The GLP-1 ELISA was performed according to the manufacturer’s protocol.
